# Therapeutic Effect of *Garcinia cambogia* Extract and Hydroxycitric Acid Inhibiting Hypoxia-Inducible Factor in a Murine Model of Age-Related Macular Degeneration

**DOI:** 10.3390/ijms20205049

**Published:** 2019-10-11

**Authors:** Mari Ibuki, Chiho Shoda, Yukihiro Miwa, Ayako Ishida, Kazuo Tsubota, Toshihide Kurihara

**Affiliations:** 1Laboratory of Photobiology, Keio University School of Medicine, Tokyo 160-8582, Japan; shirayuki0727@yahoo.co.jp (M.I.); syouda.chiho@nihon-u.ac.jp (C.S.); yukihiro226@gmail.com (Y.M.); ayakoishida9696@keio.jp (A.I.); 2Department of Ophthalmology, Keio University School of Medicine, Tokyo 160-8582, Japan; 3Department of Ophthalmology, Nihon University, Tokyo 101-8309, Japan

**Keywords:** hydroxycitric acid, *Garcinia cambogia*, age-related macular degeneration, laser induced neovascularization, hypoxia-inducible factor, retina, choroid

## Abstract

Background: Age-related macular degeneration (AMD) is the leading cause of blindness and can be classified into two types called atrophic AMD (dry AMD) and neovascular AMD (wet AMD). Dry AMD is characterized by cellular degeneration of the retinal pigment epithelium, choriocapillaris, and photoreceptors. Wet AMD is characterized by the invasion of abnormal vessels from the choroid. Although anti-vascular endothelial growth factor (VEGF) therapy has a potent therapeutic effect against the disease, there is a possibility of chorio-retinal atrophy and adverse systemic events due to long-term robust VEGF antagonism. We focused on hypoxia-inducible factor (HIF) regulation of VEGF transcription, and report the suppressive effects of HIF inhibition against ocular phenotypes in animal models. Many of the known HIF inhibitors are categorized as anti-cancer drugs, and their systemic side effects are cause for concern in clinical use. In this study, we explored food ingredients that have HIF inhibitory effects and verified their effects in an animal model of AMD. Methods: Food ingredients were screened using a luciferase assay. C57BL6/J mice were administered the *Garcinia cambogia* extract (Garcinia extract) and hydroxycitric acid (HCA). Choroidal neovascularization (CNV) was induced by laser irradiation. Results: Garcinia extract and HCA showed inhibitory effects on HIF in the luciferase assay. The laser CNV model mice showed significant reduction of CNV volume by administering Garcinia extract and HCA. *Conclusions:* Garcinia extract and HCA showed therapeutic effects in a murine AMD model.

## 1. Introduction

Age-related macular degeneration (AMD) is the leading cause of blindness worldwide [1,2]. AMD is known as a chronic disease in ophthalmology. In addition to a genetic background, various environmental factors have been reported to be involved with AMD onset [3]. Large randomized controlled trials have shown that consumption of antioxidants, such as vitamin C, vitamin E, and zinc, suppressed the progression of AMD [4].

AMD can be classified into two types: atrophic AMD (dry AMD) and neovascular AMD (wet AMD). Dry AMD is characterized by cellular degeneration of the retinal pigment epithelium (RPE), choriocapillaris, and photoreceptors. Wet AMD is characterized by the invasion of abnormal vessels from the choroid into the outer retinal layers, which causes leakage, a hemorrhage, and fibrosis. In diabetic retinopathy (DR) and wet AMD, abnormal blood vessel growth is largely caused by a vascular endothelial growth factor (VEGF) [5,6,7]. Therefore, intravitreal administration of anti-VEGF drugs has been established and achieved therapeutic results [8]. Choroidal atrophy and side effects on other organs due to long-term administration have also been reported [9,10]. Thus, a novel safer and less invasive approach needs to be developed.

We previously focused on hypoxia-inducible factors (HIFs), which are transcriptional factors that are stabilized and activated under hypoxic conditions [11]. HIFs consist of α- and β-subunits that bind hypoxia responsive elements (HRE) in target genes as heterodimers. Under normoxic conditions, α-subunits of HIFs, HIF-αs, are hydroxylated by prolyl hydroxylase, ubiquitinated by von Hippel-Lindau (VHL) protein, and degraded step-by-step. Under hypoxic conditions, the activity of HIF-α prolyl hydroxylase decreases, which results in HIF-α stabilization [12]. At least three different α-subunits of HIFs exist including HIF-1α, HIF-2α, and HIF-3α. Among them, HIF-1α is expressed ubiquitously in most cell types and is known to control various genes such as the *Vegf*, BCL2 interacting protein 3 (*Bnip3*), and phosphoinositide-dependent kinase 1 (*Pdk1*) [13]. By inhibiting HIFs, we reported that the volume of choroidal neovascularization (CNV) is reduced in a wet AMD animal model known as a laser-induced CNV mouse model [14]. We also reported that HIF inhibitors suppressed angiogenesis in an oxygen-induced retinopathy model [15]. Most known HIF inhibitors are anti-cancer agents [16,17], and they cause various side effects [18]. Therefore, a clinical application is difficult. We think that searching for a safer and less invasive HIF inhibitor and examining the effect in animal models will lead to the development of a new therapeutic method.

*Garcinia cambogia* extract (Garcinia extract) is extracted from *G. cambogia* fruit, which is eaten raw in Asia. Garcinia extract is known to contain abundant hydroxycitric acid (HCA), which is a derivative of citric acid. This is effective for weight loss [19] and already used as a dietary supplement. HIF regulates cellular homeostasis including metabolism [20], and we hypothesized that Garcinia extract has an HIF inhibitory effect. Recently, we reported that administering an HIF inhibitor, which was screened using a luciferase assay, has a therapeutic effect in animal models [21,22].

In this study, we examined the HIF inhibitory effect of Garcinia extract and its main component, HCA. We further evaluated the therapeutic effect of these two substances in a murine model of laser-induced CNV.

## 2. Results

### 2.1. HIF Activation Suppressed by Garcinia Extract and HCA Administration in Vitro

The murine retinal cone cell line (661W) and the human RPE cell line (ARPE19) were used to evaluate HIF activity with a luciferase assay since photoreceptors and RPE cells significantly contribute the pathogenesis of AMD even though organoids or differentiated cells derived from iPS cells of AMD patients may be considered for better in vitro systems. Under a hypoxic condition, the activity of HIFα prolyl hydroxylase (PHD) decreases, which results in HIFα stabilization [12]. CoCl_2_ was added to stabilize the inhibition of PHD [23] and to activate HIF signaling. Chetomin was used as a positive control of the HIF inhibitor. We used Garcinia extract. Table 1 lists its components showing that HCA accounts for more than half of the extract. Garcinia extract and HCA showed an HIF inhibitory effect compared with the control group in ARPE19 cells (Figure 1A) and 661W cells (Figure 1B).

### 2.2. Administration of Garcinia Extract and HCA Downregulated Hif1a and Downstream Genes

We examined how Garcinia extract and HCA affect mRNA expression of *Hif1a* and the downstream genes. In ARPE19 cells, *Hif1a* was significantly downregulated by administration of Garcinia extract regardless of the presence or absence of CoCl_2_ (Figure 2A). The downstream genes of HIFs such as *Vegfa, Bnip3,* and *Pdk1,* were upregulated by CoCl_2_ and significantly downregulated by Garcinia extract administration (Figure 2B–D). Similarly, *Hif1a* was downregulated by administration of Garcinia extract in 661W cells (Figure 2E). CoCl_2_-induced upregulation of *Vegfa* was also downregulated by Garcinia extract administration in 661W cells (Figure 2F). Expression of other downstream genes of HIFs showed a tendency to be downregulated as well as *Vegfa* (Figure 2G,H). HCA also downregulated *Hif1a* and the downstream genes in ARPE19 cells (Figure 3A–D) and 661W cells (Figure 3E–H). Both Garcinia extract and HCA suppressed HIF-1α protein expression increased by CoCl_2_ administration in ARPE19 cells (Figure 4A,B) and 661W cells (Figure 4C,D).

### 2.3. Administration of Garcinia Extract Suppressed CNV Volume in the Model Mice

An MF diet mixed with Garcinia extract at a concentration of 0.2% was administered to the mice for seven weeks. The laser was irradiated six weeks after beginning the administration. The volume of CNV was evaluated on the seventh day after irradiation. A significant reduction in the volume of CNV was observed in the Garcinia extract group compared with the control group (Figure 5A,B).

### 2.4. Administration of HCA Suppressed CNV Volume in the Model Mice

HCA suspended in corn oil was injected intraperitoneally at 30 mg/kg/day for a total of two weeks, and the mice were irradiated with a laser one week after beginning the injections. The volume of CNV on the seventh day of the irradiation was significantly reduced in the HCA administration group when compared with the control group (Figure 6A,B).

### 2.5. Administration of HCA Suppressed HIF-1α Expression in Vivo

HCA suspended in corn oil was intraperitoneally administered (30 mg/kg/day) to the mice for a total of 10 days, and the mice were irradiated with a laser on the seventh day of administration. In the retina and the choroid of the mice on the third day of irradiation, HIF-1α increased due to the laser irradiation and suppressed due to the administration of HCA (Figure 7A,B) even though the signal with the RPE/choroid tissue was weak.

## 3. Discussion

In this study, we employed a luciferase assay to assess the HIF inhibitory effects of Garcinia extract and its major ingredient, HCA. Administration of substances screened by the luciferase assay suppressed CNV formation in a murine model. The data show that the luciferase assay is suitable for screening substances with potentially therapeutic effects in in vivo models. As a transcriptional factor, the activity of HIF-α can be inhibited at the level of transcription, translation, protein degradation, and DNA binding [15]. In this study, we showed Garcinia extract and HCA suppressed mRNA expression of *Hif1a,* which indicates that these agents inhibit HIF activity at a transcription level (Figure 2 and Figure 3).

Although intravitreal injection of anti-VEGF drugs has been established to be the first line treatment of CNV in AMD, pathological myopia, and macular edema, in DR and retinal vein occlusion, the anti-VEGF treatment is still invasive. Atrophic thinning of the retina has been reported to be a result of long-term administration, and the cost to patients is also high [9,10]. Therefore, development of a less invasive and safer treatment is needed.

HIF plays an important role in maintaining cellular homeostasis in response to changes in the oxygen status [11]. Angiogenesis is one of the most major hypoxia responses that can be mediated by the HIF/VEGF axis [13,24]. To date, several studies have been conducted to reveal the effect of HIF inhibition on ocular neovascularization [14,15,25,26]. Previously, we demonstrated that RPE-specific HIF knockout mice showed a significant reduction in laser-induced CNV volume when compared with VEGF knockout mice [14]. Clustered regularly interspaced short palindromic repeat (CRISPR)-mediated HIF-1α inactivation could also suppress CNV formation in mice [27]. Pharmacological intervention such as administration of digoxin, aloe-emodin, or topotecan suppresses retinal angiogenesis by inhibiting HIF-1α and its downstream pathways including VEGF [15,26,28]. These results indicate that HIF inhibition is a promising approach for managing ocular neovascularization.

HCA, which is the main component of Garcinia extract, is similar in chemical structure to citric acid, and, thus, inhibits the action of adenosine triphosphate (ATP) citrate lyase in the citric acid cycle [29,30]. This action inhibits the conversion of citric acid to acetyl coenzyme A (CoA) and suppresses fatty acid synthesis. The increased amount of citric acid that is not converted to acetyl CoA leads to acceleration of glycogen production from glucose. Thus, the intake of HCA stabilizes the blood sugar level, which results in the suppression of the hunger sensation. Therefore, a preventive effect against hyperphagia is also expected [31,32,33]. In chicken hepatocytes, HCA decreased the accumulation of lipid droplets and accelerated energy metabolism [34]. HCA protected the cells from ER stress by increasing the antioxidant status and mitochondrial functions [35]. We showed that HCA suppressed CNV formation by suppressing HIF expression (Figure 6 and Figure 7). Anti-inflammatory and anti-tumor effects of HCA have previously been reported [36]. The HIF inhibitory effect of HCA may also induce these phenotypes.

Traditionally, fruits of *Garcinia cambogia* have been eaten in Asia. Additionally, Garcinia extract and HCA have been widely distributed in the world and used as a dietary supplement. Thus, their safety has been established and they can be adapted as preventive medicine when compared with other inhibitors for either VEGF or HIF.

## 4. Materials and Methods

### 4.1. Animals

All animal experiments were performed in accordance with the National Institutes of Health (NIH) guidelines for work with laboratory animals and the ARVO Animal Statement for the Use of Animals in Ophthalmic and Vision Research and Animal Research: Reporting In Vivo Experiments (ARRIVE) guidelines. All animal experiments were approved by the Institutional Animal Care and Use Committee at Keio University (#16017-(2) on 12 October 2018). C57BL6/J mice were purchased from CLEA Japan (Tokyo, Japan) and were raised in standard transparent mouse cages in an air-conditioned room maintained at 23 ± 3 °C under a 12-h dark/light cycle with free access to food and water.

### 4.2. Luciferase Assay

We performed a luciferase assay using 661W and ARPE19, which were both transfected HIF-luciferase reporter gene constructs (Qiagen, Hilden, Netherlands). These constructs encode the firefly luciferase gene under the control of HRE, which bind HIFs as previously described [15,21]. As an internal control, these cells were co-transfected with a CMV-renilla luciferase construct. We seeded cells in 0.8 × 10^4^ cells/well/70 μL in HTS Transwell^®^-96 Receiver Plate, White, TC-Treated, Sterile (Corning, Corning, NY, USA). At 24 h after seeding, HIF-αs were induced by 200 μM CoCl_2_ (FUJIFILM Wako Pure Chemical, Osaka, Japan). Garcinia extract (*Garcinia Cambogia* Extract 50% (Table 1), BIO ACTIVES JAPAN, Tokyo, Japan) and HCA (FUJIFILM Wako Pure Chemical) were dissolved in dimethyl sulfoxide (DMSO; FUJIFILM Wako Pure Chemical, Osaka, Japan) and added into the growth medium at the same time as CoCl_2_. We added each compound dissolved in DMSO to the cell medium so that its concentration was 1 mg/mL considering the toxicity of the material [37]. After the administration, cells were incubated for 24 h at 37 °C in a 5% CO_2_ incubator. Quantitation of the luciferase expression was performed using the Dual-Luciferase^®^ Reporter Assay System (Promega, Madison, WI, USA). The fluorescent intensity was read by a microplate reader (Biotek, Winooski, VT, USA). Additionally, 100 nM of chetomin (Sigma-Aldrich Japan, Tokyo, Japan) was used as a positive control for an HIF inhibitor and a DMSO-containing medium was used without CoCl_2_, Garcinia extract, and HCA as a vehicle control.

### 4.3. Laser-Induced CNV

The mice’s eyes were dilated with 0.5% tropicamide and 0.5% phenylephrine eyedrops (Santen Pharmaceutical, Osaka, Japan) and anesthetized with a combination of midazolam (Sandoz, Tokyo, Japan), medetomidine (Orion, Espoo, Finland), and butorphanol tartrate (Meiji Seika Pharma, Tokyo, Japan), called MMB. The laser procedure was performed as previously reported [38]. To focus on the retina, the eyes of mice were covered with a contact lens (Haag-Streit Diagnostics, Köniz, Switzerland). In each eye, five 532-nm argon laser spots (200 mW, 100 ms, 75 mm) were placed between the retinal vessels, located 2-disc diameters from the optic nerve head. When irradiating with the laser, the air bubble was used as an index of Bruch’s membrane disruption. Laser lesions that lacked this air bubble or that were hemorrhaged were excluded from the data analysis.

### 4.4. CNV Volume Measurement

On the seventh day after the irradiation, mice were sacrificed, and the eyes were enucleated. The choroid-sclera complex was flat-mounted and fixed in 4% paraformaldehyde solution for 1 h. After washing, the tissue was stained with isolectin B4 (Invitrogen, Carlsbad, CA, USA) at 4 °C for 3 days. After encapsulation, CNV was observed with a laser microscope (Zeiss, Oberkochen, Germany). Three-dimensional images of CNV were generated using ZEN software 2011 (Zeiss) and the volume was measured using Imaris® (Bitplane, Zurich, Switzerland).

### 4.5. Administration of Garcinia Extract and HCA to Mice

An MF diet (Oriental Yeast, Tokyo, Japan) mixed with Garcinia extract at a concentration of 0.2% was administered to 4-week-old male mice for a total of 7 weeks while considering the toxicity of the material [37]. The control group was administered an MF diet. The mice were irradiated with a laser 6 weeks after beginning administration. We injected 30 mg/kg/day HCA suspended in corn oil intraperitoneally to 6-week-old male mice 5 days/week for a total of 2 weeks. The control group was injected with corn oil. The laser was irradiated 1 week after the initial injection.

### 4.6. Real-Time PCR (Polymerase Chain Reaction)

We extracted RNA from the ARPE19 cell line using a TRI reagent® (MRC, Cincinnati, OH, USA) and an Econospin column for RNA (GeneDesign, Osaka, Japan). The columns were washed with Buffer RPE (Qiagen, Hilden, Netherlands) and Buffer RWT (Qiagen, Hilden, Netherlands). RNA was reverse-transcribed into cDNA using ReverTra Ace® qPCR RT Master Mix with gDNA remover (TOYOBO, Osaka, Japan) [21]. Real-time-PCR was performed using THUNDERBIRD® SYBR® qPCR Mix (TOYOBO, Osaka, Japan) with a StepOnePlus Real-Time PCR System (Applied Biosystems, Foster City, CA, USA). The relative amplification of cDNA fragments was calculated using the 2^−ΔΔ*C**t*^ method. Real-time PCR primer sequences were as follows: *Vegf* forward: TCTACCTCCACCATGCCAAGT, *Vegf* reverse: GATGATTCTGCCCTCCTCCTT, *Glut1* forward: CGGGCCAAGAGTGTGCTAAA, *Glut1* reverse: TGACGATACCGGAGCCAATG, *Pdk1* forward: ACAAGGAGAGCTTCGGGGTGGATC, *Pdk1* reverse: CCACGTCGCAGTTTGGATTTATGC, *Bnip3* forward: GGACAGAGTAGTTCCAGAGGCAGTTC, *Bnip3* reverse: GGTGTGCATTTCCACATCAAACAT, *Gapdh* forward: TCCCTGAGCTGAACGGGAAG, and *Gapdh* reverse, GGAGGAGTGGGTGTCGCTGT.

### 4.7. Western Blot

For in vitro experiments, we added 200 μM CoCl_2_, 1 mg/mL Garcinia extract, and HCA to ARPE19 cell line while considering the toxicity of the material [37]. Six hours after the administration, cells were collected in the RIPA buffer (Thermo Fisher Scientific, Waltham, MA, USA) and mixed with protease inhibitors (Roche Diagnostics, Basel, Switzerland) and MG132 (Cayman Chemical, Ann Arbor, MI, USA). Then, the cells were homogenized. Afterward, we centrifuged the samples (14,800 rpm, 4 °C, 30 min) and collected the supernatant. The protein concentration was adjusted to 75 μg/30 μL.

For in vivo experiments, mice were sacrificed, and eyes were enucleated on the third day after the laser irradiation. We pooled 6 ocular samples from 3 mice per group. Retinas and choroids were placed into a lysis buffer (10 mmol/L Tris-HCl, 100 mmol/L NaCl, 1 mmol/L EDTA, and 1% Triton X-100), mixed with protease inhibitors and MG132, and were homogenized. Then, we centrifuged the samples (14,800 rpm, 4 °C, 30 min) and collected the supernatant. The protein concentration was adjusted to 55 μg/42 μL.

The following procedures were similarly performed for both in vitro and in vivo experiments. After adjusting the protein concentration, each sample was fractionated using 10% SDS-PAGE and transferred to polyvinylidene fluoride (PVDF, Merck, Darmstadt, Germany) membranes. The membranes were blocked with 5% skim milk for 1 h at room temperature and incubated at 4 °C with rabbit monoclonal antibodies against HIF-1α (1:1000, Cell Signaling Technology, Danvers, MA, USA), or mouse monoclonal antibodies against β-actin (1:10,000, Cell Signaling Technology). After washing, the membranes were incubated for 1 h at room temperature with a 1:3000 dilution of horseradish peroxidase (HRP, GE Healthcare, Chicago, IL, USA)-labeled secondary antibody for HIF-1α, or with a 1:10,000 dilution of HRP-labeled secondary antibody for β-actin. The immunoreactive signal was detected using EzWestLumi plus (Atto, Tokyo, Japan) and protein bands were visualized with a chemiluminescence (ImageQuant LAS 4000 mini, GE Healthcare, Chicago, IL, USA). Blots were quantified by ImageJ (National Institutes of Health, Bethesda, MD, USA).

### 4.8. Statistics

A two-tail Student’s *t*-test for comparison of 2 groups and ANOVA-Tukey for the comparison of 3 or 4 groups were used. We considered probability values less than 0.05 as being statistically significant. All results are expressed as the mean ± standard deviation.

## 5. Patents

ROHTO Pharmaceutical Co. Ltd. has applied for a patent for the therapeutic effect of the Garcinia extract and HCA.

## Figures and Tables

**Figure 1 ijms-20-05049-f001:**
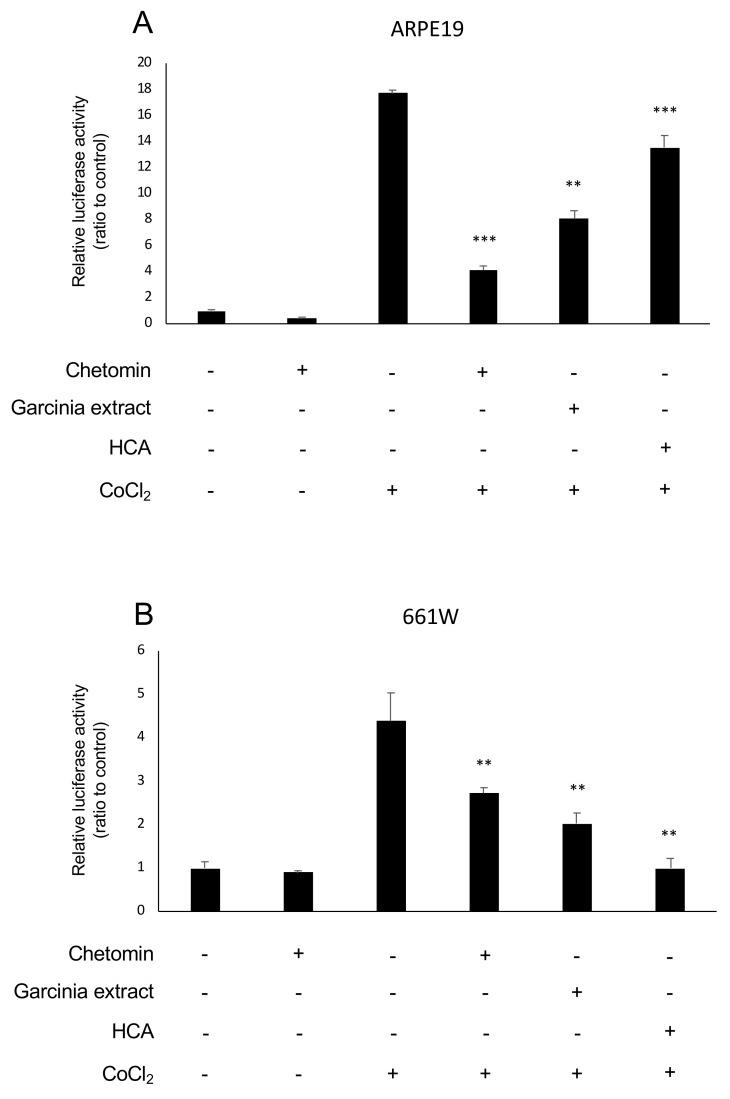
*Garcinia cambogia* extract (Garcinia extract) and hydroxycitric acid (HCA) suppressed hypoxia-inducible factors (HIF) activation in vitro. (**A**) Administration of Garcinia extract and HCA significantly suppressed CoCl_2_-induced HIF activation in ARPE19 cells. (**B**) Administration of Garcinia extract and HCA significantly suppressed CoCl_2_-induced HIF activation in 661w cells. ** *p* < 0.01, *** *p* < 0.001 compared with CoCl_2_ without chetomin, Garcinia extract, and HCA, *n* = 3.

**Figure 2 ijms-20-05049-f002:**
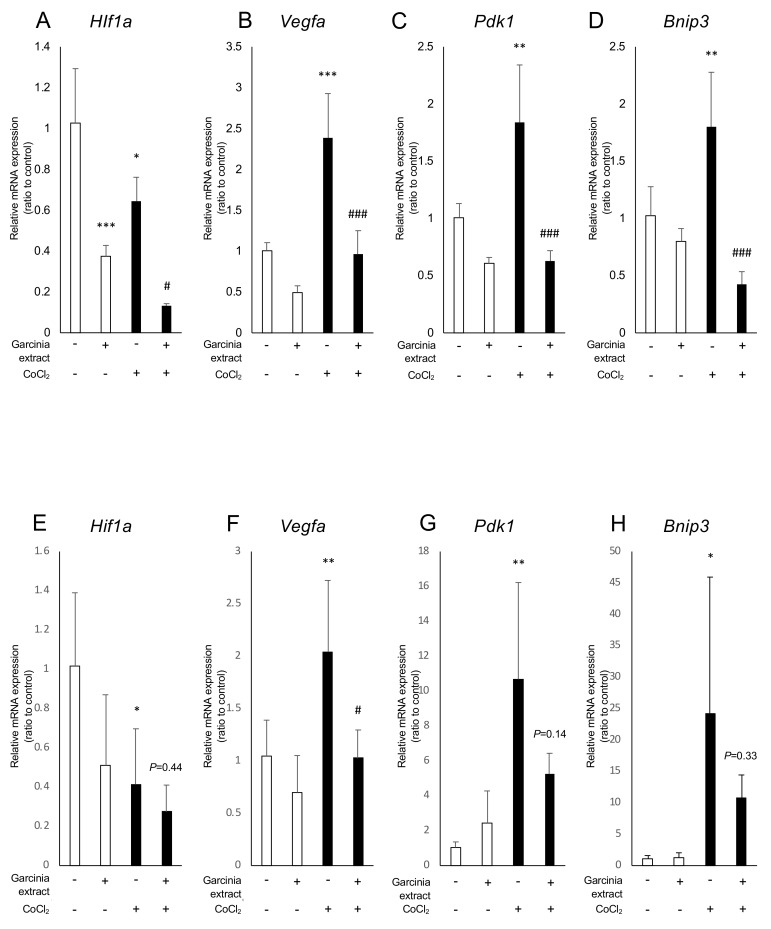
*Hif1α* and the downstream genes were downregulated by Garcinia extract administration. (**A**) *Hif1a* was downregulated by Garcinia extract administration with or without CoCl_2_ in APRE19 cells. The downstream genes of HIFs, including (**B**) *Vegfa*, (**C**) *Pdk1*, and (**D**) *Bnip3* were significantly downregulated by the administration of Garcinia extract in ARPE19 cells. (**E**) *Hif1a* was downregulated by Garcinia extract administration in 661W cells significantly without CoCl_2_. (**F**) *Vegfa* was significantly downregulated by the administration of Garcinia extract in 661W cells. (**G**) *Pdk1* and (**H**) *Bnip3* also showed a similar tendency. * *p* < 0.05, ** *p* < 0.01, *** *p* < 0.001 compared with the control. ^#^
*p* < 0.05, ^###^
*p* < 0.001 compared with CoCl_2_ without chetomin and Garcinia extract, *n* = 3–6.

**Figure 3 ijms-20-05049-f003:**
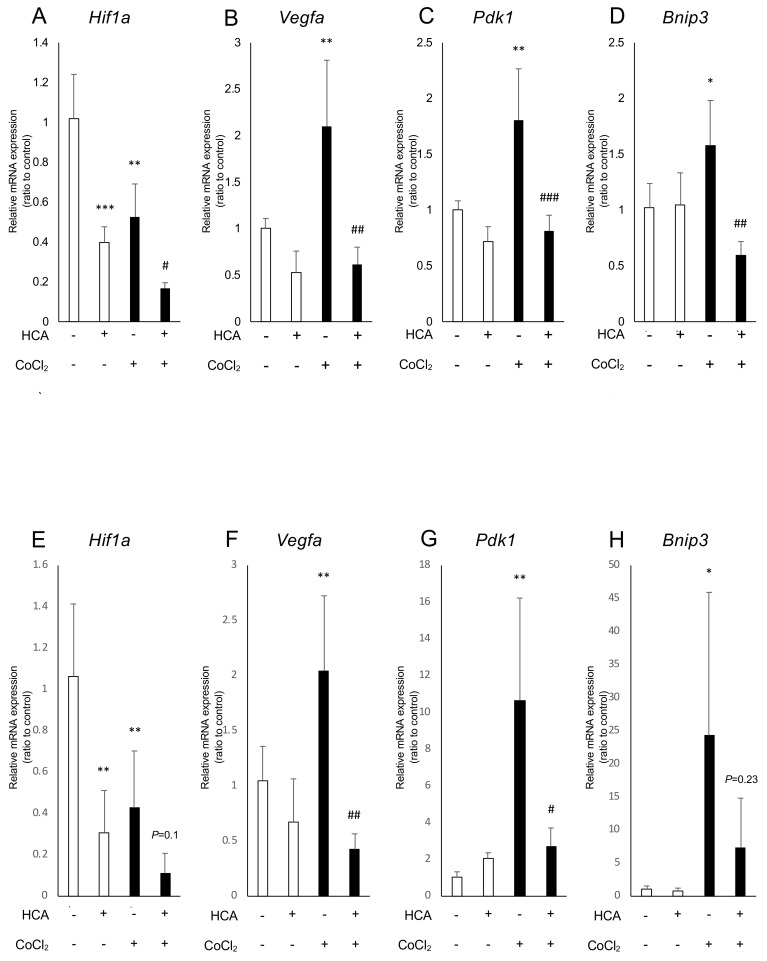
*Hif1a* and the downstream genes were downregulated by HCA administration. (**A**) *Hif1a* was downregulated by HCA administration with or without CoCl_2_ in ARPE19 cells. The downstream genes of HIFs including (**B**) *Vegfa*, (**C**) *Pdk1*, and (**D**) *Bnip3* were significantly downregulated by administration of HCA in ARPE19 cells. *Hif1a* was downregulated by HCA administration in 661W cells significantly without CoCl_2_. (**F**) *Vegfa* and (**G**) *Pdk1* were significantly downregulated by administration of HCA in 661W cells. (**H**) *Bnip3* also showed a similar tendency. * *P* < 0.05, ** *p* < 0.01, *** *p* < 0.001 compared with the control. ^#^
*p* < 0.05, ^##^
*p* < 0.01, ^###^
*p* < 0.001 compared with CoCl_2_ without chetomin and HCA, *n* = 3–6.

**Figure 4 ijms-20-05049-f004:**
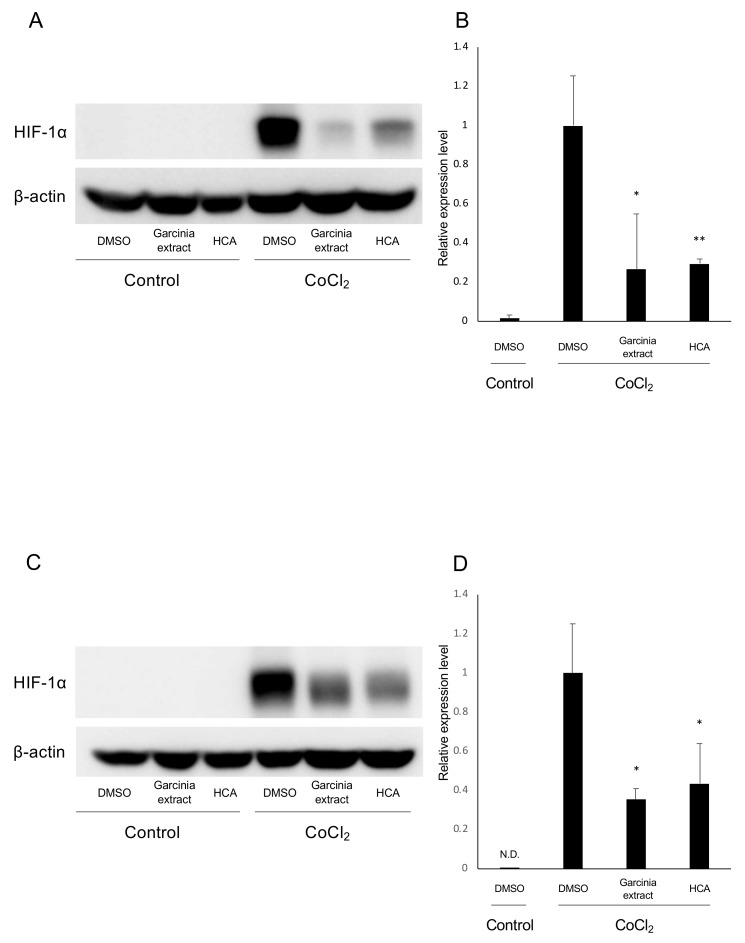
HIF-1α protein expression is suppressed by Garcinia extract and HCA administration in vitro. (**A**) Western blot for HIF-1α and β -actin in ARPE19 cells. (**B**) Quantification of the blots showed that the administration of Garcinia extract and HCA significantly suppressed increased HIF-1α protein expression in ARPE19 cells (*n* = 3). (**C**) Western blot for HIF-1α and β-actin in 661W cells. (**D**) Quantification of the blots showed that the administration of Garcinia extract and HCA significantly suppressed increased HIF-1α protein expression in 661W cells (*n* = 3). N.D.: not detectable. * *p* < 0.05, ** *p* < 0.01.

**Figure 5 ijms-20-05049-f005:**
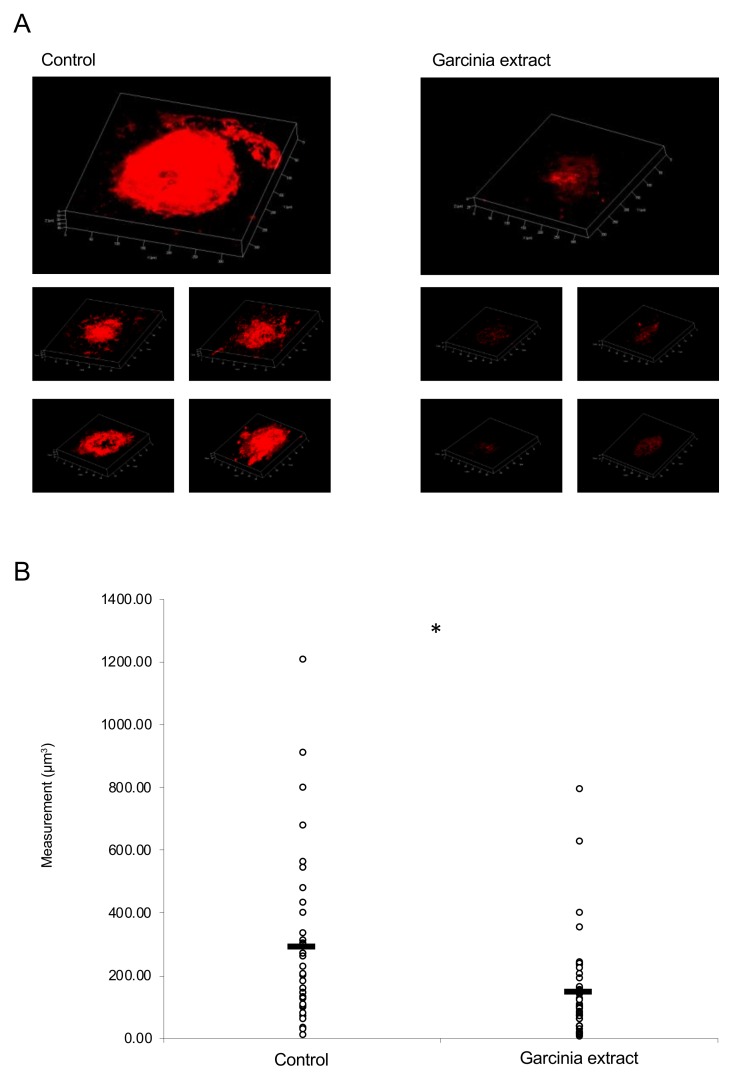
Administration of Garcinia extract suppressed choroidal neovascularization (CNV) volume in mice. (**A**) Representative three-dimensional images of isolectin B4 (IB4)-stained CNV. Individual CNV from each mouse was shown. (**B**) Quantification of the laser-induced CNV volume. Note that the administration of Garcinia extract significantly reduced the CNV volume when compared with the control group. Control: 248.09 ± 262.52 μm^3^, Garcinia extract: 151.58 ± 152.96 μm^3^, *n* = 44 for each, 5 mice for each, * *p* < 0.05.

**Figure 6 ijms-20-05049-f006:**
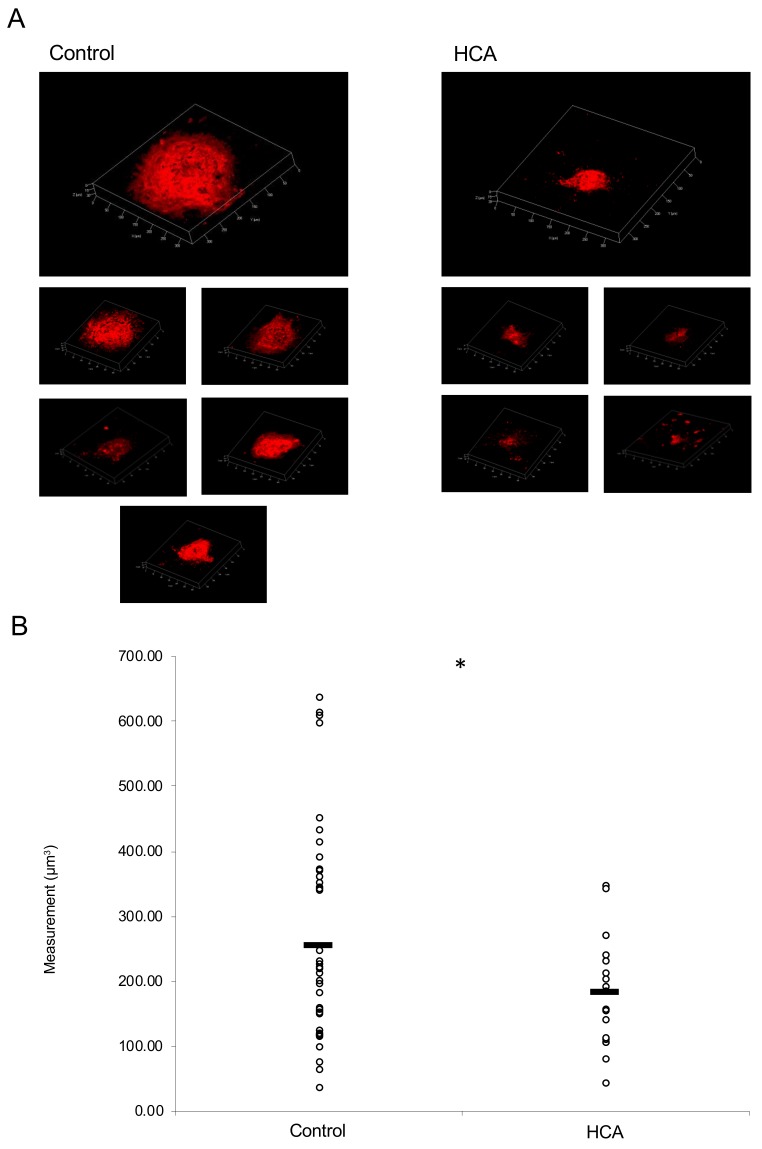
Administration of HCA suppressed CNV volume in mice. (**A**) Representative three-dimensional images of IB4-stained CNV. Individual CNV from each mouse was shown. (**B**) Quantification of the laser-induced CNV volume. Note that the administration of HCA significantly reduced the CNV volume when compared with the control group. Control: 278.05 ± 186 μm^3^, HCA 195.87 ± 107.57 μm^3^, *n* = 25, and 44 for control and HCA, respectively. Six and five mice for control and HCA, respectively. Control group: 25 CNV, six mice, HCA: 44 CNV, five mice, * *p* < 0.05.

**Figure 7 ijms-20-05049-f007:**
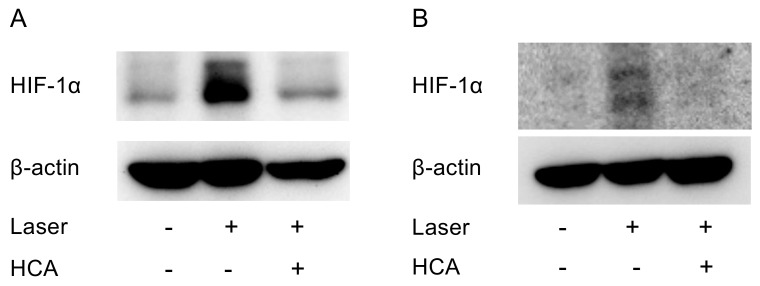
HIF-1α protein expression is suppressed by HCA administration in vivo. Western blot for HIF-1α with the tissue samples from (**A**) the retina or (**B**) the retinal pigment epithelium (RPE) /choroid. Note that administration of HCA suppressed HIF-1α expression, which increased due to the laser irradiation in the posterior ocular segments.

**Table 1 ijms-20-05049-t001:** Specification of the *Garcinia cambogia* Extract 50%.

Parameter	Specification Result
Main Characteristics	
Description	Pale brown color with characteristic odor
Identification	Compliance with IR (infrared spectroscopy) spectrum
Loss on drying	2.87% *w/w*
Solubility	Slightly soluble in hot water and insoluble in water
**Particle Information**	
pH (1% *w/v* solution)	9.7
Tapped bulk density	0.61 g/mL
Loose bulk density	0.36 g/mL
Sieve test (mesh size)	100% passed with 60 mesh
Active Ingredients	Results
HCA	56.55% *w/w*
Calcium by titration	19.70% *w/w*
**Contaminants**	
Heavy Metals	<10 ppm
Lead	1.0 ppm
Arsenic	<0.1 ppm
Cadmium	<0.01 ppm
Mercury	0.1 ppm
**Microbiological profile**	
Total microbial plate count	600 CFU/g
Total yeast and mold count	10 CFU/g
*Escherichia coli*	Absent
Coliforms	Absent
Salmonella	Absent
*Staphylococcus aureus*	Absent
*Pseudomonas aeruginosa*	Absent

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
