# Peer review of "Therapeutic Effect of Garcinia cambogia Extract and Hydroxycitric Acid Inhibiting Hypoxia-Inducible Factor in a Murine Model of Age-Related Macular Degeneration"

_ijms, 2019, doi:10.3390/ijms20205049_

Round 1
Reviewer 1 Report
In this paper, the authors demonstrated the therapeutic effects of Garcinia extract and hydroxycitric acid (HCA) on the mouse AMD model through inhibiting hypoxia-inducible factor (HIF).
There are some minor issues that need to be addressed by the authors.
How did the dosages of Garcinia extract determined (50% in vitro and 0.2% in vivo)? In the animal experiment, why the units of dose are different (0.2% of Garcinia extract and 30 mg/kg of HCA)? For the Western blot of HIF expression in vivo (Fig. 7), how many times did the authors repeat the experiment? It will be more convincing if the authors show the statistical analysis of Western bolt quantification from repeated experiments. HIF expressions from at least 3 mice per group are recommended to be tested. The same issue exists in Fig. 4. Have the author tested the HIF expression of retina/RPE/choroid from mice treated with Garcinia extract? Does dose Garcinia extract have any inhibitory effect on HIF expression in vivo? It is better to write "3days" as "3 days". Similar issues can be found in many sentences in the paper.Author Response
Please see the attachment.

Reviewer 2 Report
The manuscript „Hydroxycitric acid inhibiting hypoxia-inducible factor has a therapeutic effect in a murine model of age-related macular degeneration“ by Ibuki et al. reports on some aspects of inhibition of the hypoxia-inducible factor alpha (HIF) regulating VEGF transcription. Specifically, the authors want to identify natural food compounds which might be suitable as novel therapeutic agents for treating choroidal neovascularization in late stage AMD.
While the authors address a topic of medical interest in AMD research, the manuscript fails to meet adequate standards for scientific writing and diligence, specifically with regard to methodological procedures, scientific context and, most annoying, with regard to greatly flawed language due to epidemic spelling errors and grammatical inadequacies throughout the text.
Comments:
Abstract: Late stage AMD is a more complex disease than presented by the authors. Importantly, AMD is not simply “characterized with pathological angiogenesis in the retina”. Geographic atrophy which may occur by its own or in combination with neovascular complications is another important aspect of AMD disease, but is entirely ignored by the authors. Abstract: Wording and scientific content is inadequate for a scientific summary. Line 34: “…is one of the major leading causes of blindness…” reads awkward and should be rephrased accordingly. Line 37: “…have been reported to be involved with the onset.” This is an inaccurate representation of scientific and importantly epidemiological research. The parameters listed are suspected to be a risk factor (not “involved with the onset”), although not every factor quoted has been reproduced in major studies conducted some years ago. Line 44: what is meant by “DR”? Any abbreviation should be introduced when mentioned first, then the abbreviation should be used throughout the text. Line 49: When introducing hypoxia-inducible factors family members, subunits and how these factors interact in coordinating transcription regulation should be briefly reviewed. Line 56: Reference [14] refers to a targeted deletion mouse model and NOT to a laser-induced CNV mouse model. Line 57: Reference [15] refers to a murine oxygen-induced retinopathy model. This should be clarified. Line 58: Are the anti-cancer drugs simply “… expected to cause various side effects” or is there any scientific evidence for it? Lines 61-66: Here is the place to broadly introduce Garcinia extracts and its possible importance for VEGF inhibition. The discussion in lines 174-189 of the discussion would then be repetitive and thus could be omitted. Line 72: The choice of 661W and ARPE19 cells seems random but need to be clarified in more detail. Also, it needs to be explained whether there are possibly better cell line models to reflect the situation in an AMD patient. Line 73: “…to evaluate HIF activity…”. Exactly which HIF factor was examined? Line 77: “…we verified HCA which is a main component of Garcinia extract…”. Why was HCA chosen specifically and not any other component of Garcinia extracts? This needs to be clarified and (negative?) results with other components of Garcinia extracts need to be shown. Line 81: Figure 1; Line 87: Figure 2; Line 104: Figure 3. It is unclear to which reference the significance levels relate to. Line 146: Figure 7B: The quality of the HIF1a Western Blot is inadequate to make a useful point. Throughout the text: Gene and protein symbols/names should be given according to scientific standards. Line 216-217: The luciferase assay is inappropriately described with reference to publication [14]. When checking this reference there is referral to yet another publication with only little more but inadequate explanation on the assay. This is not tolerable as this assay is the method of choice in the current manuscript and thus is crucial to be understood in its details. Line 220 and elsewhere: It is unclear if appropriate vehicle controls (e.g. DMSO) were used or if cells not treated with Garcinia extract or HCA were left completely untreated. Results: The two cell lines were used for luciferase and qRT PCR experiments, although HIF1alpha protein expression was only validated by Western Blot of ARPE19 samples. This is inconsistent. Why excluded the authors any 661W results at this point? Results: Numbers of animals used for experiments should be stated in the respective figure legends. Lines 174-185: Please see comment #10. Line 224: It is unclear exactly what was positively and what was negatively controlled?Author Response
Please see the attachment.

Reviewer 3 Report
The authors should satisfactorily address the following points in order to improve the quality of the manuscript and better demonstrate the strength of their results.
The title should contain both: Garcinia extract and hydroxycitric acid (HCA) since both were used side by side throughout the paper and in the conclusions the authors mention both. FIG 1: The authors should not repeat the same result twice. Specifically, 1rst , 2nd & 4th bars in FIG 1A and 1C as well as 1B and 1D are the same data, the only difference within each pair of graphs is in the samples treated with Garcinia extract (3rd Bars) or HCA (5th Bars). The bar graphs need to be consolidated. qPCR results in the presence of Garcinia extract (FIG 2) and HCA (FIG 3) should be combined in a single figure to maintain consistency with FIG 1. The authors should mention the rational for using CoCl2 to induce HIF1a-dependent responses. FIG 5-6: The authors need to describe better how these figures were done and show that the lesion depicted as representative is so. Specifically: the authors should show instead of one selected lesion the complete fundus image comparing eyes from treated and control mice as shown in the reference 42, FIG 4, i.e. the paper cited by the authors to reference the CNV-laser method, they used. Also, in material and methods it should be clearly informed how many laser lesions, the authors made, per retina? How many animals were used in each of these two figures? FIG 7. How many eyes were pulled together to prepare the lysate? How many times this WB was repeated? How many ug of proteins were loaded per lane? It is very surprising to have such a profound impact in the back of the eye with a drug orally administered. Is the Hif1a response suppressed in other tissues throughout the body as well?
Round 2
Reviewer 3 Report
The authors have answered all the points of my first review.
This manuscript is a resubmission of an earlier submission. The following is a list of the peer review reports and author responses from that submission.